# Predictive Factors of Giant Cell Arteritis in Polymyalgia Rheumatica Patients

**DOI:** 10.3390/jcm11247412

**Published:** 2022-12-14

**Authors:** André Ramon, Hélène Greigert, Paul Ornetti, Jean-Francis Maillefert, Bernard Bonnotte, Maxime Samson

**Affiliations:** 1Rheumatology Department, Dijon-Bourgogne University Hospital, 21000 Dijon, France; 2INSERM, EFS BFC, UMR 1098, RIGHT Graft-Host-Tumor Interactions/Cellular and Genetic Engineering, Bourgogne Franche-Comté University, 21000 Dijon, France; 3Internal Medicine and Clinical Immunology Department, Dijon-Bourgogne University Hospital, 21000 Dijon, France; 4Vascular Medicine Department, Dijon-Bourgogne University Hospital, 21000 Dijon, France; 5INSERM, CIC 1432, Clinical Investigation Center, Plurithematic Module, Technological Investigation Platform, Dijon-Bourgogne University Hospital, 21000 Dijon, France; 6INSERM UMR 1093-CAPS, UFR des Sciences et Du Sport, Bourgogne Franche-Comté University, 21000 Dijon, France

**Keywords:** polymyalgia rheumatica, giant cell arteritis, predictive factor, subclinical, biomarkers

## Abstract

Polymyalgia rheumatica (PMR) is an inflammatory rheumatism of the shoulder and pelvic girdles. In 16 to 21% of cases, PMR is associated with giant cell arteritis (GCA) that can lead to severe vascular complications. Ruling out GCA in patients with PMR is currently a critical challenge for clinicians. Two GCA phenotypes can be distinguished: cranial GCA (C-GCA) and large vessel GCA (LV-GCA). C-GCA is usually suspected when cranial manifestations (temporal headaches, jaw claudication, scalp tenderness, or visual disturbances) occur. Isolated LV-GCA is more difficult to diagnose, due to the lack of specificity of clinical features which can be limited to constitutional symptoms and/or unexplained fever. Furthermore, many studies have demonstrated the existence—in varying proportions—of subclinical GCA in patients with apparently isolated PMR features. In PMR patients, the occurrence of clinical features of C-GCA (new onset temporal headaches, jaw claudication, or abnormality of temporal arteries) are highly predictive of C-GCA. Additionally, glucocorticoids’ resistance occurring during follow-up of PMR patients, the occurrence of constitutional symptoms, or acute phase reactants elevation are suggestive of associated GCA. Research into the predictive biomarkers of GCA in PMR patients is critical for selecting PMR patients for whom imaging and/or temporal artery biopsy is necessary. To date, Angiopoietin-2 and MMP-3 are powerful for predicting GCA in PMR patients, but these results need to be confirmed in further cohorts. In this review, we discuss the diagnostic challenges of subclinical GCA in PMR patients and will review the predictive factors of GCA in PMR patients.

## 1. Introduction

PMR is an inflammatory rheumatism characterised by arthromyalgia of the shoulder and pelvic girdles [1] associated with an elevation of acute phase reactants. In most cases, PMR is isolated; however, it can also reveal GCA, as 16 to 21% of patients with PMR exhibit features of giant cell arteritis (GCA) in temporal artery biopsy [2]. GCA is the most common vasculitis occurring in patients over 50 years old. It is a granulomatous vasculitis that affects large-size arteries [3]. In 40 to 60% of cases it is associated with polymyalgia rheumatica (PMR) [4]. Diagnosis of GCA is based (1) on a range of clinical and biological features that built a pre-test probability of GCA and (2) on a “proof” of vasculitis that can be brought by vascular imaging or by temporal artery biopsy (TAB) [5,6]. GCA must be diagnosed promptly because this allows initiation of glucocorticoids treatment which reduces the risk of ischemic complications, in particular visual ones [7]. Two GCA phenotypes can be distinguished: cranial GCA (C-GCA) and large vessels-GCA (LV-GCA). These two phenotypes may be isolated or associated in 29 to 83% of cases [8,9].

In clinical practice, the main challenge is identifying LV-GCA in patients with PMR, given that clinical signs of LV-GCA are less specific because they are often limited to constitutional symptoms (asthenia, anorexia, weight loss, unexplained fever) with an increase in acute phase reactants (erythrocyte sedimentation rate (ESR) and C-reactive protein (CRP)). Signs of vascular insufficiency, such as upper-extremity asymmetric blood pressure, abolition of peripheral radial/pedal pulses, upper/lower limb claudication, aortic regurgitation to cardiac auscultation, and vascular bruit in the carotids and limb arteries are much more suggestive of large vessel vasculitis but are less common [10,11,12,13]. The presence of undiagnosed vascular damage of the aorta exposes patients to an increased risk of complications, particularly aneurysms and/or aortic dissection, which rarely reveal the disease, as they typically occur after several months or years of evolution [13,14,15]. By contrast, cranial GCA features, such as temporal headaches, scalp tenderness, jaw claudication, and visual disturbances, are very suggestive of GCA; therefore, it is not an issue in routine clinical practice to identify C-GCA in a patient with PMR [10]. The frequent association between PMR and GCA raises the question of ruling out GCA in a patient presenting with PMR features, since this could lead to adjustments in the dosage of glucocorticoids (GC) or initiation of tocilizumab in patients with an increased risk of GC side effects [7]. However, performing expensive and/or invasive complementary exams to rule out GCA in the case of isolated PMR does not seem possible for all patients, given that PMR is three to six times more common than GCA [16]. Numerous studies have demonstrated that a various proportion of patients with isolated PMR features have subclinical GCA. This association highlights the need to identify the predictive factors of GCA in PMR patients in order to identify patients for whom imaging or TAB are relevant, and ultimately reducing the risk of ischemic events within this population.

Herein, we propose to discuss the occurrence of subclinical GCA in PMR patients and review the predictive factors of GCA in a population of PMR patients.

## 2. The Difficult Issue of Subclinical GCA in Patients with Apparently Isolated PMR

Most retrospective studies have reported significant rates of subclinical GCA in patients with apparent isolated PMR. A recent meta-analysis reported a pooled prevalence of subclinical GCA in new onset PMR of 23% [17]. Gonzales-Gay et al. [18], based on a study of 210 biopsy-proven GCA patients, showed that 11 patients (5.2%) had no clinical signs of GCA at diagnosis and during the first year of follow-up. Schmidt et al. [19] prospectively evaluated temporal arteries with colour-doppler ultrasound (CDUS) and demonstrated that 7% of patients with an apparent isolated PMR showed evidence of vasculitis (halo sign, stenosis, and/or occlusion). Similarly, several studies have also demonstrated the presence of histological evidence of GCA in 4 to 41% of TAB from patients with apparently isolated PMR [20,21,22,23,24]. These results are difficult to analyse, as they come from different studies with different methodological biases, and because the results also depend on the sensitivity of the clinician. However, this demonstrates that even if a patient only has PMR features and no cephalic signs of GCA, there may still be subclinical temporal artery involvement, which may lead to a formal diagnosis of GCA.

The increasing use of [18F] fluorodeoxyglucose (FDG)-positron emission tomography (PET)/computed tomography (CT) in GCA and PMR patients has also shown that patients with apparently isolated PMR exhibit significant vascular uptake and are in fact GCA patients. The incidence of this subclinical GCA ranges from 10 to 92% depending on the studies [25,26,27,28]; thus, these wide incidence variations raise questions about patient selection and how the interpretation of vascular examinations may be standardised in these studies (qualitative versus quantitative [18F] FDG-PET/CT analysis, for instance). However, several investigators have suggested that GCA patients with isolated PMR features at baseline belong to a benign subset of GCA patients with a low risk of developing a GCA-related ischemic event. In a prospective study focusing on isolated PMR, Blockmans et al. [29] found subclinical GCA in 31% of patients using [18F] FDG-PET/CT. Interestingly, this study showed that these patients with PMR and subclinical GCA features did not experience more relapses than PMR patients without GCA during the first year of follow-up after treatment onset at a PMR dosage (i.e., 15 mg/day of prednisone). In contrast to these reassuring data, other retrospective studies have reported higher rates of ischemic complications, ranging from 14 to 27%, in PMR patients with subclinical GCA during follow-up [30,31,32]. Due to the retrospective design of these studies, it is difficult to know whether PMR patients who developed GCA had subclinical GCA at baseline or whether they developed GCA during the course of the disease. To definitively answer this question, prospective case-control studies are needed.

## 3. Are There Clinical Predictive Factors of GCA in PMR Patients?

In a patient with PMR symptoms, the occurrence of cranial signs (temporal headaches, temporal artery abnormalities, scalp tenderness, jaw claudication, and/or visual disturbances) directly leads the clinician to suspect GCA, and this can be confirmed by performing TAB [6] and/or CDUS [7]. In a multivariate analysis focusing on clinical features of GCA in a population of patients with PMR, Rodriguez-Valverde et al. [33] showed that the clinical signs which were most predictive of a diagnosis of GCA were headaches (Odds Ratio (OR) = 13.6), temporal artery abnormalities (OR = 4.2), and jaw claudication (OR = 4.8). Fairly similar results were obtained in another study [34]. However, the main challenge for the clinician is not diagnosing C-GCA, but rather detecting subclinical GCA, i.e., usually GCA with large vessel involvement (LV-GCA). Hemmig et al. identified inflammatory back pain and absence of lower limb pain as being associated with subclinical LV-GCA in apparently isolated PMR patients. However, no correlation was found between inflammatory back pain and FDG uptake in the abdominal aorta or in bursae of the lumbar spinous processes [17]. Glucocorticoid resistance could be a sign of associated GCA during the course of PMR. From a small sample of glucocorticoid-resistant PMR patients, Cimmino et al. [35] found features of LV-GCA after they performed [18F] FDG-PET/CT in 37.5% of patients.

## 4. PMR and GCA Pathogenesis: Clues to Identified Specifics Disease Biomarkers

PMR and GCA share some pathophysiological features, including a common immunogenetic background with a high frequency of HLA-DRB1 (HLA-DRB1*0401 and HLA-DRB1*0404 alleles) [36,37]. GCA is strongly associated with the presence of the HLA-DRB1*0701 allele, which is not present in PMR [37]. The pathophysiological “thread” of PMR currently remains poorly understood, with most findings based on the study of circulating cells. A possible pathophysiological continuum between PMR and GCA was demonstrated by Ma-Krupa et al. [38]. The authors demonstrated the presence of dendritic cells (DCs) expressing maturation markers such as CD83 and CCR7 in TABs of PMR patients. However, these mature DCs were not associated with inflammatory infiltration of the arterial wall. Several studies have also assessed serum concentrations of cytokines involved in the pathogenesis of PMR and GCA. Compared with control patients, patients with GCA or PMR have higher serum levels of IL-6, CXCL-9, CXCL-10, and BAFF [39,40]. As in GCA, increased serum IL-6 and T-cell polarization toward Th1 and Th17 pathways associated with decreased CD4 regulatory T cells (CD4^+^CD25^high^FOXP3^+^) have been described in PMR [41,42]. In GCA, IL-6 is produced by macrophages and vascular smooth muscle cells in the arterial wall of affected arteries [43]. By definition, there is no vasculitis in PMR and the source of IL-6 could be macrophages infiltrating the synovial tissue (of the shoulder bursae) [44]. As the main difference between isolated PMR and GCA is the inflammation of the vascular wall, the identification of biomarkers involved in angiogenesis/vascular remodelling seems relevant for discriminating between these two pathologies.

Serum biomarkers could be used to identify GCA (Table 1) in patients with isolated PMR for whom performing imaging and/or TAB would be the most appropriate.

### 4.1. Elevation of Acute Phase Reactant

Both GCA and PMR are characterised by an increase in acute phase reactants, such as ESR and CRP. It is commonly accepted that GCA is suspected in a PMR patient with a significant increase in ESR and/or CRP [48]. Some authors have demonstrated a risk of GCA in PMR patients with ESR > 100 mm/h, elevated liver enzymes, platelet count, and anaemia [33,34,49]. In a retrospective study, Fukui et al. [46] also found higher ESR values in patients with isolated GCA and GCA/PMR overlap versus patients with isolated PMR. Van Sleen et al. [47] reported data from a cohort of GCA/PMR overlap (*n* = 10 patients) and isolated PMR patients (*n* = 29) in which they assessed the ability of CRP and ESR to discriminate between these two conditions. The area under the curve (AUC) for CRP and ESR was <0.8, which is not accurate enough to allow the clinician to use only these data to distinguish between GCA and PMR. Our group found no significant difference in serum IL-6 levels between GCA and PMR patients [42]. Similarly, Van Sleen et al. showed that serum IL-6 was not accurate enough to distinguish between GCA/PMR overlap and isolated PMR (AUC = 0.57) [47]. Furthermore, there is clear evidence that GCA patients with cranial ischemic complications (mainly visual ischemic events) have significantly lower inflammatory markers and serum IL-6 levels than GCA patients without ischemic complications. This highlights the risk of using CRP or ESR alone to rule out GCA diagnosis in a patient with PMR [50].

### 4.2. Vascular Remodelling Markers

Rather than markers of systemic inflammation, the study of markers revealing vascular aggression or remodelling may be more relevant, as these processes are more specific to vasculitis and should therefore be absent in isolated PMR. Van Sleen et al. [45,47] have identified several potential biomarkers for discriminating between GCA/PMR overlap and isolated PMR. An elevated angiopoietin-2 (a protein involved in angiogenesis) level was predictive of GCA/PMR overlap. The AUC for angiopoietin-2 was 0.90 which seems to make it a powerful marker for ruling out GCA in a PMR patient. The angiopoietin-2/angiopoietin-1 ratio can also predict GCA in PMR patients with an AUC of 0.88 in the UMCG cohort (cut-off value > 0.051) and an AUC of 0.78 in the Aarhus cohort (cut off value > 0.048). The same authors also demonstrated the accuracy of metalloproteinase 3 (MMP-3) for identifying GCA in PMR patients. They found that a low level of MMP-3 was predictive of GCA (AUC = 0.81 in the Aarhus cohort (threshold < 23 ng/mL) and AUC = 0.82 in the UMCG cohort (threshold < 14 ng/mL)). The low level of MMP-3 during GCA could be explained by the consumption of MMP-3 due to the production of higher levels of MMP-9, which is known to be implicated in vascular remodelling [51]. These results are consistent with the study by Fukui et al. [46], who also found a higher level of MMP-3 in isolated PMR patients compared to GCA/PMR overlap. However, there is a major difference in MMP-3 cut-off values between these two studies, indicating the need to standardise these assay techniques before testing them in daily practice.

## 5. The Role of Imaging in Identifying GCA

Imaging currently plays a major role in the diagnosis of GCA. The 2018 European Alliance of Associations for Rheumatology (EULAR) recommendations for imaging in large vessels vasculitis [5] suggest performing CDUS as first-line imaging for the assessment of cranial arteries. For the assessment of extracranial arteries, it is recommended to use either CDUS (with a limited value for aortitis assessment), magnetic resonance imaging (MRI), [18F] FDG-PET/CT, or computed tomography angiography (CTA) (Table 2) [5]. Systematic imaging in a patient with isolated PMR is not currently recommended.

### 5.1. Ultrasound (US) Imaging

US is the first-line examination for evaluating cranial arteries in suspected GCA, and in some centres it tends to replace TAB as the gold standard for GCA diagnosis [5]. Its inclusion into a clinical fast-track has significantly reduced the occurrence of ischemic events in patients with GCA [61,62]. The two most important features suggestive of GCA are the halo sign (Figure 1) and the compression sign [63]. When compared to clinical diagnosis, US of the temporal arteries has shown high diagnostic accuracy for C-GCA. A meta-analysis showed a pooled sensitivity and specificity of 77 and 96%, respectively [52]. The inclusion of stenosis and occlusion in addition to the halo sign slightly increases sensitivity but decreased specificity (78 and 89%, respectively). Compared to TAB, pooled sensitivity and specificity were 68 and 81%, respectively [60].

It has also been shown that evaluating the axillary arteries as well as the temporal arteries increased the sensitivity of US without altering its specificity [56,64]. More recently, the diagnostic performance of ultrasound for diagnosis of LV-GCA (excluding aortitis) has been demonstrated. The authors compared a “standard” ultrasound workup (evaluation of the distal temporal and axillary arteries) with an “extended” workup (addition of the study of the large supra-aortic vessels). The “extended” ultrasound identified 93 patients with LV-GCA, compared to 56 with the “standard” ultrasound, which showed the relevance of exploring additional arterial segments with ultrasound [65].

US has the advantage of being easily available, non-irradiating, and able to assess the cranial and extra-cranial arteries (carotid, axillary, and subclavian arteries). Further studies have reported a high inter- and intra-reliability rate for the halo sign and compression sign assessments [63,66,67]. For practitioners with no previous experience in vascular US, the training program increased the sensitivity of US for GCA diagnosis [66,68]. However, US is less sensitive for assessment of the aorta [5], which can be the only large artery affected in some GCA patients and can be responsible for very few symptoms other than constitutive signs and/or PMR.

### 5.2. [18F] FDG-PET/CT

The [18F] FDG-PET/CT has demonstrated a high degree of accuracy for diagnosing LV-GCA. Several studies have investigated its diagnostic performance and reported sensitivity ranging from 61 to 80% and specificity from 79 to 100%. [53,57,58,59]. Recent studies have also demonstrated that C-GCA can be detected effectively using [18F] FDG-PET/CT [53,54,69,70]. Our group has also demonstrated that a combination of cranial and extracranial FDG-PET/CT improved overall sensitivity (80%) for GCA diagnosis through its ability to capture both phenotypes (LV-GCA and C-GCA) with a single exam (Figure 2) [53]. The main limitations of FDG-PET/CT are related to the poorer performance of this exam in cases of diabetes and the reduction of its sensitivity after starting treatment with glucocorticoids. The normally accepted time frame is 3 days for standard FDG-PET; however, this is not known with any precision for cephalic PET [71]. Imfeld et al. [72] compared the diagnostic performance of CDUS and standard [18F] FDG-PET/CT in a prospective study and concluded that these exams were complementary. Indeed, standard [18F] FDG-PET/CT allows for better exploration of the aorta and ultrasound allows for a better evaluation of the cranial arteries. Combining these two examinations increased the diagnostic yield of GCA from 16 to 20%. From a validation PMR cohort for FDG-PET/CT scores, the authors have demonstrated that FDG-PET uptakes (Leuven score and Besancon score (mean)) were lower in PMR/GCA overlap in comparison with isolated PMR patients. No correlation with clinical or laboratory findings were found [73].

Prieto-Peña et al. [25] identified some predictive factors of positive [18F] FDG-PET/CT for LV-GCA in 84 patients with isolated PMR that met the ACR/EULAR 2012 classification criteria. The main predictive factors of large vessel vasculitis were the persistence of PMR features despite low-dose glucocorticoid treatment (15 mg/day), the occurrence of an increase in acute phase reactants, constitutional symptoms (fever, asthenia, weight loss), and the occurrence of atypical clinical signs, such as inflammatory back pain (OR = 4.7 (1.03; 21.5)) or diffuse bilateral lower limb pain (OR = 8.8 (1.70; 46.3)). Additionally, new PET radiotracers that target cells implicated in the pathogenesis of GCA and PMR (macrophages, T cells, endothelial cells) are currently being studied [74] and may provide a better understanding of the links between these two conditions.

### 5.3. MRI and CTA

MRI is used to assess cranial arteries demonstrating arterial wall thickness and the enhancement of cranial arteries in cases of GCA. A recent meta-analysis that involved 10 MRI studies in C-GCA demonstrated a pooled sensitivity and specificity of 75 and 89%, respectively, when compared to clinical diagnosis. When compared to TAB, sensitivity and specificity increased to 91 and 78%, respectively [55]. Better diagnostic performance for the evaluation of wall thickening and mural enhancement in GCA patients was also demonstrated for fat-supressed 3D high-resolution T1-weighted black-blood MRI (CUBE T1) compared to 2D contrast-enhanced vessel-wall MRI [75]. The advantage of using 3-Dimensional MRI is its multiplanar reconstructions, which are useful when studying extra- and intra-cranial arteries [76].

Few studies have prospectively compared the accuracy of MRI with CDUS in GCA patients. Yip et al. demonstrated that US was more sensitive than MRI for detecting a change in supra-aortic large vessels, especially in patients with longstanding diseases (defined as an active disease diagnosed at least 6 months before inclusion in the study). No differences were found between MRI and CDUS for cranial arteries assessment [77]. However, its accessibility remains the main limitation in the context of a diagnostic emergency, although this should not delay the administration of glucocorticoids.

Few studies have reported good diagnostic accuracy for CTA. In one study [57], the authors reported a sensitivity of 73% and a specificity of 78% for the diagnosis of LV-GCA. Our group defined an aortic wall thickening threshold of ≥2.2 mm in favour of GCA [78]. The main limitation of CTA is related to the use of iodinated contrast medium and irradiation.

## 6. Conclusions

For PMR patients, the occurrence of cranial features leads the clinician directly to suspect GCA. In this situation, a colour-doppler ultrasound of the temporal arteries is very efficient, especially if performed by a trained operator. It is therefore recommended by EULAR as a first-line test to confirm diagnosis, especially if the pre-test probability of GCA is high. 

In clinical practice, the difficulty arises from the possibility that patients with apparently isolated PMR may in fact have associated large-vessel vasculitis, particularly aortitis. This type of association is often asymptomatic, but discovering it may lead to changing the dosage of glucocorticoid treatment to prevent long-term complications (aneurysm or dissection). It is however difficult in daily practice to detect aortitis through imaging. Ultrasound, which is non-irradiating and easier to access, can help by studying the axillary arteries and supra-aortic trunks. This review shows that the best features for suspected subclinical vasculitis in a patient with PMR are persistent inflammatory syndrome under low-dose glucocorticoids treatment (15 mg/day), glucocorticoid resistance, or clinical signs that evoke involvement of the arteries of the upper or lower limbs.

Some biomarkers, in particular MMP-3 and angiopoietin-2, seem promising; however, these data need to be confirmed through larger cohorts and dosing methods require standardisation. The development of biomarkers to distinguish patients with isolated PMR from those with overlapping PMR/GCA should therefore be prioritised to improve the efficacy and cost- of managing these patients. These markers must be very sensitive to select the patients for whom a TAB or/and imaging test should be performed, depending on the technical platform available in each centre.

Due to its accessibility, CDUS is undoubtedly the easiest test, but it must be performed by a trained operator. If the test is negative and there is still a strong clinical suspicion of LV-GCA, a PET-FDG/CT with cranial and extra-cranial arteries assessment seems to us to be the most efficient test to date.

## Figures and Tables

**Figure 1 jcm-11-07412-f001:**
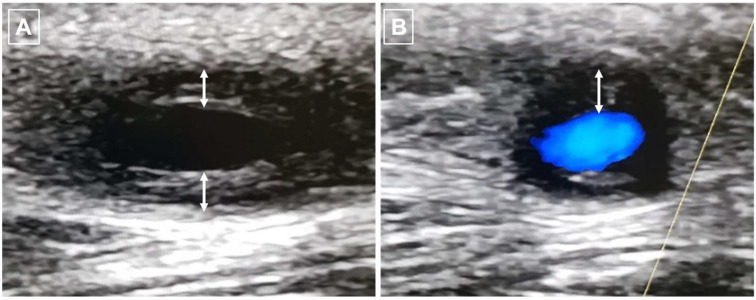
Ultrasound of the temporal artery with halo sign characterised by parietal thickening (double arrow) in B mode (**A**) and colour-doppler (**B**) leading to a reduction in arterial lumen.

**Figure 2 jcm-11-07412-f002:**
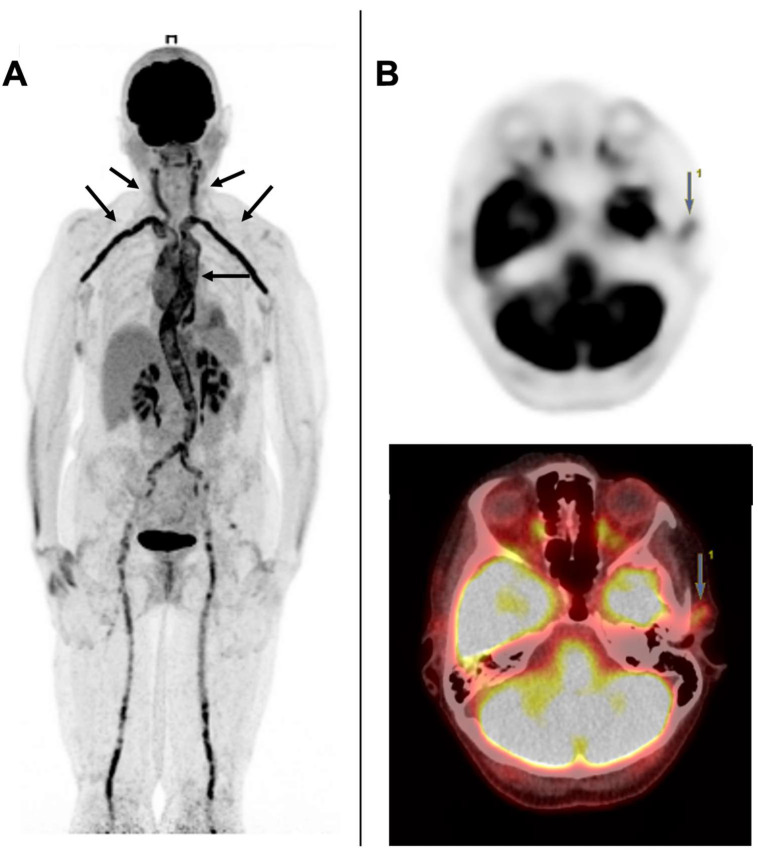
[18F] FDG-PET/CT in giant cell arteritis (GCA). Panel (**A**) shows large vessel vasculitis of subclavian, carotid arteries and aorta (black arrows). Panel (**B**) shows tracer uptake in the left temporal artery (arrow) after dedicated cephalic acquisition.

**Table 1 jcm-11-07412-t001:** Potential biomarkers for discriminating GCA between PMR.

Biomarker	Study Design	Population	AUC	Threshold	References
ESR	Prospective 2 international cohorts (Aarhus, UMCG)	GCA/PMR overlap versus Isolated PMR	Aarhus: 0.82 UMCG: 0.77	Aarhus: 60 mm/h UMCG: 91 mm/h	[45]
	Retrospective	GCA/PMR overlap versus Isolated PMR	NR	NR	[46]
Angiopoietin 2	Prospective UMCG cohort	GCA/PMR overlap versus Isolated PMR	0.90	3124 pg/mL	[47]
Angiopoietin 2/Angiopoeitin 1 ratio	Prospective 2 international cohorts (Aarhus, UMCG)	GCA/PMR overlap versus Isolated PMR	Aarhus: 0.78 UMCG: 0.88	Aarhus: 0.048 UMCG: 0.051	[45]
MMP-3	Prospective 2 international cohorts (Aarhus, UMCG)	GCA/PMR overlap versus Isolated PMR	Aarhus: 0.81 UMCG: 0.82	Aarhus: 23 ng/mL UMCG: 14 ng/mL	[45]
	Retrospective	GCA/PMR overlap versus Isolated PMR	0.81	140 ng/mL	[46]

AUC: Area Under the Curve; ESR: erythrocyte sedimentation rate; MMP3: metalloproteinase 3; GCA: giant cell arteritis; PMR: polymyalgia rheumatica; UMCG: University Medical Center Groningen.

**Table 2 jcm-11-07412-t002:** Imaging in C-GCA and LV-GCA: classical features, diagnostic accuracy, advantages, and disadvantages.

		C-GCA			LV-GCA		
		US	[18F] FDG-PET/CT	MRI	US	[18F] FDG-PET/CT	CTA
Features		Halo sign Compression sign Stenose Occlusion	Vascular uptake Vascular occlusion and stenosis	Mural thickening Enhancement of cranial arteries	Halo sign Stenose Occlusion	Vascular uptake Vascular occlusion and stenosis	Mural thickening Arteries enhancement
Diagnostic accuracy	Compared to clinical diagnosis	Se: 77% Spe: 96% [52]	Se: 71-73.3% Spe: 91-97.2% [53,54]	Se: 75% Spe: 89% [55]	Se: 100% Spe: 91% [56] *	Se: 61-80% Spe: 79-100% [53,57,58,59]	Se:73% Spe: 78% [57]
	Compared to TAB	Se: 68% Spe: 81% [60]	Se: 92% Spe: 85% [54]	Se: 91% Spe: 78% [55]			
Advantages		No radiation Fast track clinics Low cost High resolution Availability	Overview of involved arteries Detection of GCA/PMR mimickers (infection, neoplasia)	No radiation No iodinated contrast agents	No radiation Fast track clinics Low cost High resolution Availability	Overview of involved arteries Detection of GCA/PMR mimickers (infection, neoplasia)	Good overview of the aorta Fast acquisition
Disadvantages		Trained operator (especially for large vessels assessment)	Radiation (25 mSv) High cost Completion time Decrease in sensitivity after 3 days of glucocorticoids	High cost Low availability Limited by metal devices, claustrophobia, or pacemaker	Limited value for aortitis assessment Trained operator (especially for large vessels assessment)	Radiation (25 mSv) High cost Completion time Decrease in sensitivity after 3 days of glucocorticoids	Radiation (17mSv) Iodinated contrast medium

C-GCA: cranial giant cell arteritis; LV-GCA: large vessels giant cell arteritis; US: ultrasound; [18F] FDG-PET/CT: [18F] fluorodeoxyglucose positron emission tomography/computed tomography; MRI: magnetic resonance imaging; CTA: computed tomography angiography; TAB: temporal artery biopsy; Se: sensitivity; Spe: specificity; PMR: polymyalgia rheumatica; *: diagnostic performance of CDUS with the evaluation of axillary arteries in combination to temporal arteries.

## Data Availability

Not applicable.

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
