# Peer review of "Predictive Factors of Giant Cell Arteritis in Polymyalgia Rheumatica Patients"

_jcm, 2022, doi:10.3390/jcm11247412_

Round 1

Reviewer 1 Report

This is a well-written review on the difficulties in discriminating patients with isolated PMR from patients with overlapping GCA, which has important clinical implications. Most of the important studies have been mentioned and the overall storyline makes sense

àGeneral comments

-The formatting of the paragraphs could be altered, although this might be a personal preference. The paragraphs are often very short and sometimes only one sentence. This makes the manuscript appear somewhat like a list of findings rather than a storyline. Combining storyline under one topic sentence in a paragraph might improve that somewhat.

-Could the authors speculate somewhat more how the pathogenesis compares and differs between both diseases? Of course, few studies exist on PMR biopsies but a few have been published, and blood studies have more often been published. These differences might provide clues on what to look for in a disease-specific biomarker.

àSpecific comments

-Maybe you could add in the introduction that besides the higher GC dose in GCA, GCA patients also could qualify for tocilizumab treatment, which has not been approved for PMR.

-In table 1 CRP is named as a potential marker of overlapping GCA. However, the stated AUCs are so low that they are likely meaningless. I would suggest to leave CRP out here. Same goes for IL-6.

-“clear evidence that GCA patients with ischemic complications have significantly lower inflammatory markers and serum IL-6 levels”
This is an important point. To my knowledge however, most evidence only shows this association for visual ischemic complications.

-The sentences 156-160 could be taken out as they indeed do not investigate differences between GCA/PMR. The sentences 161-163 could than be added to the paragraph on acute-phase reactants.

-A possible interesting finding that could be discussed that PMR activity on PET may be lower in patients that have overlapping GCA: van der Geest et al, doi: 10.1093/rheumatology/keab483

-An important meta-analysis is missing: Hemmig et al, doi: 10.1016/j.semarthrit.2022.152017

-Final sentence: “a study of the temporal arteries”; this means a TAB of US?

Author Response

Reviewer 1:

We thank the reviewers for the time spent reviewing our paper and for the suggestions to improve its quality.

A point-by-point reply that indicates how the manuscript has been revised is provided below. Changes in the text appear in red.

All authors have reviewed the manuscript, all agree with its contents, and have approved its submission.

We sincerely hope that these revisions have made the manuscript more acceptable for publication in Journal of Clinical Medicine.

General comments

  • The formatting of the paragraphs could be altered, although this might be a personal preference. The paragraphs are often very short and sometimes only one sentence. This makes the manuscript appear somewhat like a list of findings rather than a storyline. Combining storyline under one topic sentence in a paragraph might improve that somewhat.

Answer:

Thank you for this comment. We have modified the manuscript formatting with removing non necessary paragraphs.

  • Could the authors speculate somewhat more how the pathogenesis compares and differs between both diseases? Of course, few studies exist on PMR biopsies but a few have been published, and blood studies have more often been published. These differences might provide clues on what to look for in a disease-specific biomarker.

Answer:

According to this comment, we have added a new paragraph about PMR and GCA pathogenesis similarities and difference.

Changes in text: line 131

  1. PMR and GCA pathogenesis: Clues to identified specifics disease biomarkers

PMR and GCA share some pathophysiological features, including a common immunogenetic background with a high frequency of HLA-DRB1 (HLA-DRB1*0401 and HLA-DRB1*0404 alleles) [38,39]. GCA is strongly associated with the presence of the HLA-DRB1*0701 allele which is not present in PMR [39].

The pathophysiological "thread" of PMR currently remains poorly understood, with most findings based on the study of circulating cells. A possible pathophysiological continuum between PMR and GCA was demonstrated by Ma-Krupa et al. [40]. The authors demonstrated the presence of dendritic cells (DCs) expressing maturation markers such as CD83 and CCR7 in TABs of PMR patients. However, these mature DCs were not associated with inflammatory infiltration of the arterial wall.  Several studies have also assessed serum concentrations of cytokines involved in the pathogenesis of PMR and GCA. Compared with control patients, patients with GCA or PMR have higher serum levels of IL-6, CXCL-9, CXCL-10 and BAFF [41,42]. As in GCA, increased serum IL-6 and T-cell polarization toward Th1 and Th17 pathways associated with decreased CD4 regulatory T cells (CD4+CD25highFOXP3+) have been described in PMR [41,42]. In GCA, IL-6 is produced by macrophages and vascular smooth muscle cells in the arterial wall of affected arteries [43]. By definition, there is no vasculitis in PMR and the source of IL-6 could be macrophages infiltrating the synovial tissue (of the shoulder bursae) [44].

As the main difference between isolated PMR and GCA is the inflammation of the vascular wall, the identification of biomarkers involved in angiogenesis/vascular remodeling seems relevant to discriminate these 2 pathologies.

Specific comments

  • Maybe you could add in the introduction that besides the higher GC dose in GCA, GCA patients also could qualify for tocilizumab treatment, which has not been approved for PMR.

Answer

Thank you for this comment. We have added a sentence about Tocilizumab treatment in GCA patients.

Changes in text

Line 67: The frequent association between PMR and GCA raises the question of ruling out GCA in a patient presenting with PMR features, as this could lead to adjustments in the dosage of glucocorticoids (GC) or initiation of tocilizumab in patients with an increased risk of GC-side effects [7].

  • In table 1 CRP is named as a potential marker of overlapping GCA. However, the stated AUCs are so low that they are likely meaningless. I would suggest to leave CRP out here. Same goes for IL-6.

Answer:

As suggested, we removed, of Table 1, CRP and IL-6 datas according to their poor diagnostic performance

  • “clear evidence that GCA patients with ischemic complications have significantly lower inflammatory markers and serum IL-6 levels”

This is an important point. To my knowledge however, most evidence only shows this association for visual ischemic complications.

Answer:

Thank you for this comment. In Hernandez-Rodriguez et al. study (Elevated Production of Interleukin-6 Is Associated with a Lower Incidence of Disease-Related Ischemic Events in Patients with Giant-Cell Arteritis/ DOI: 10.1161/01.CIR.0000066907.83923.32), ischemic events reported by the authors included mainly visual ischemic complications but also (in smaller proportions) stroke (n = 1/33) and tongue ischemia (N = 2/33).

Changes in text:

Line 171: Furthermore, there is clear evidence that GCA patients with cranial ischemic complications (mainly visual ischemic events) have significantly lower inflammatory markers and serum IL-6 levels than GCA patients without ischemic complications”

  • The sentences 156-160 could be taken out as they indeed do not investigate differences between GCA/PMR. The sentences 161-163 could than be added to the paragraph on acute-phase reactants.

Answer:

As suggested, we have removed the sentencesSeveral studies have also evaluated the serum concentrations of cytokines involved in the pathogenesis of PMR and GCA. Compared to control patients, GCA and PMR patients exhibit higher serum levels of IL-6, CXCL-9, CXCL-10, and BAFF [40,41]. However, GCA and PMR patients were compared to a healthy control group without any data on GCA/PMR overlap versus isolated PMR.and addedOur group found no significant difference in serum IL-6 levels between GCA and PMR patients [42]. Similarly, Van Sleen et al. showed that serum IL-6 was not accurate enough to distinguish between GCA/PMR overlap and isolated PMR (AUC=0.57) [38]” to the paragraph on acute phase reactants.

Changes in text:

Line 168: Our group found no significant difference in serum IL-6 levels between GCA and PMR patients [52]. Similarly, Van Sleen et al. showed that serum IL-6 was not accurate enough to distinguish between GCA/PMR overlap and isolated PMR (AUC=0.57) [49].

  • A possible interesting finding that could be discussed that PMR activity on PET may be lower in patients that have overlapping GCA: van der Geest et al, doi: 10.1093/rheumatology/keab483

Answer:

Thank you for this comment. We have added this study in PET-FDG/CT paragraph. However, we didn’t find clear explanation about this lower PET-FDG uptake in GCA/PMR overlap patients.

Changes in text:

Line 258: From a validation PMR cohort for PET-FDG/CT scores, author have demonstrated that PET/FDG uptake (Leuven score and Besancon score (mean)) were lower in PMR/GCA overlap in comparison to isolated PMR patients without correlation with clinical or laboratory findings [76].

  • An important meta-analysis is missing: Hemmig et al, doi: 10.1016/j.semarthrit.2022.152017

Answer:

We have added this study in the paragraphsThe difficult issue of subclinical GCA in patients with apparently isolated PMR.” and “Are there clinical predictive factors of GCA in PMR patients?”

Changes in text:

Line 80: “A recent meta-analysis reported a pooled prevalence of subclinical GCA in new onset PMR of 23% [19].

Line 123: Hemmig et al. identified inflammatory back pain and absence of lower limb pain associated to subclinical LV-GCA in apparently isolated PMR patients. However, no correlation was found between inflammatory back pain and FDG uptake in the abdominal aorta or in bursae of lumbar spinous process [19].

  • Final sentence: “a study of the temporal arteries”; this means a TAB of US?

Answer: we apologize for this unclear sentence. By “study of the temporal arteries” we mean a study by PET-FDG (cranial and extra cranial assessment). We have corrected the sentence as following.

Changes in text:

Line 324: if the test is negative and there is still a strong suspicion of LV-GCA, a PET-FDG/CT with cranial and extra cranial arteries assessment seems to us to be the most efficient test to date.

Reviewer 2 Report

“Predictive Factors of GCA in PMR patients” is an interesting review on biomarkers and other predictive factors of GCA in PMR patients. The work is well structured and written, on a highly topical issue. The group of authors has an extensive experience on this topic. I have no major changes to report. The references and figures seem adequate and clear to me.

Author Response

We thank the reviewers for the time spent reviewing our paper and for the suggestions to improve its quality.

A point-by-point reply that indicates how the manuscript has been revised is provided below. Changes in the text appear in red.

All authors have reviewed the manuscript, all agree with its contents, and have approved its submission.

We sincerely hope that these revisions have made the manuscript more acceptable for publication in Journal of Clinical Medicine.

Response Reviewer 2:

“Predictive Factors of GCA in PMR patients” is an interesting review on biomarkers and other predictive factors of GCA in PMR patients. The work is well structured and written, on a highly topical issue. The group of authors has an extensive experience on this topic. I have no major changes to report. The references and figures seem adequate and clear to me.

Thank you for these comments and for the time spent reviewing our paper. Accordingly, to these comments no particular changes have been made.
